# First Identification and Genotyping of *Enterocytozoon bieneusi* and Prevalence of *Encephalitozoon intestinalis* in Patients with Acute Diarrhea in the Republic of Korea

**DOI:** 10.3390/pathogens10111424

**Published:** 2021-11-03

**Authors:** Ji-Young Kwon, Ji-Ye Seo, Tae-Yun Kim, Hee-Il Lee, Jung-Won Ju

**Affiliations:** Division of Vectors and Parasitic Diseases, Korea Disease Control and Prevention Agency, 187 Osongsaenmyeong2-ro, Osong-eup, Heungdeok-gu, Cheongju 28159, Chungbuk, Korea; kjiy31@korea.kr (J.-Y.K.); seojiye02@korea.kr (J.-Y.S.); kty4588@korea.kr (T.-Y.K.); isak@korea.kr (H.-I.L.)

**Keywords:** *Encephalitozoon intestinalis*, *Enterocytozoon bieneusi*, patient, genotype, phylogeny, microsporidia, acute diarrhea, Republic of Korea, prevalence rate

## Abstract

*Encephalitozoon intestinalis* and *Enterocytozoon bieneusi* can cause diarrhea in humans, especially severe diarrhea in immunocompromised patients. However, there have been few studies on *Enc. intestinalis* and *Ent. bieneusi* in patients with acute diarrhea in the Republic of Korea (ROK). In this study, fecal samples were collected from 1241 patients with acute diarrhea in 2020. Among these, 24 cases of *Enc. intestinalis* and one case of *Ent. bieneusi* were detected via PCR amplification of small subunit ribosomal RNA. Genotyping of the internal transcribed spacer region sequence revealed that the detected *Ent. bieneusi* genotype was in Group 1. This study provides the first evidence that *Ent. bieneusi* exists in humans in addition to animals in the ROK. To identify the causative agent, continuous monitoring of *Enc. intestinalis* and *Ent. bieneusi* is necessary for patients with acute diarrhea in the ROK.

## 1. Introduction

Microsporidia, which include more than 170 genera and 1300 species, are opportunistic protozoan pathogens that infect a variety of vertebrate and invertebrate hosts [1]. *Enterocytozoon, Pleistophora, Encephalitozoon, Vittaforma, Trachipleistophora, Brachiola, Nosema,* and *Microsporidium* have been reported to infect humans. *Encephalitozoon intestinalis* and *Enterocytozoon bieneusi* are known to be the most common cause of intestinal diseases [2,3]. Microsporidia are present everywhere and spread through various routes [4]. Microsporidia that infect humans have been identified in animals and water sources [5], and species infecting humans also infect animals, raising the concern for zoonotic transmission [6]. In particular, a direct evidence for zoonotic transmission of microsporidiosis has been reported in a child after close contact with *Enc. cuniculi*-infected pups [7]. In addition, spores present in stools of infected individuals can be transmitted horizontally through fecal–oral transmission or ingestion of contaminated food and water [6,8].

Microsporidia infections occur mainly in immunocompromised patients, such as patients with HIV, organ transplant recipients, and patients with cancer. Of 68 patients with HIV suffering from diarrhea in the United States, 25 and 5 were infected with *Ent. bieneusi* and *Enc. intestinalis*, respectively [9]. Of 97 patients with HIV in Germany, 18 and 2 were infected with *Ent. bieneusi* and *Enc. intestinalis*, respectively [10]. In Mali, 8 of 61 patients with HIV were infected with *Ent. bieneusi* [11]. In addition to immunocompromised patients, microsporidia infections have been continuously reported worldwide in immunocompetent individuals, including children, travelers, and the elderly [12,13]. In Mexico, 20 of 255 immunocompetent individuals were infected with *Enc. intestinalis* [14], while 39 of 1201 were infected with *Ent. bieneusi* in China [15]. Moreover, 14 of 275 pregnant women in France were infected with *Enc. intestinalis* [16]. Of 100 children with diarrhea in Gabon, 15 were infected with *Enc. intestinalis* and two with *Ent. bieneusi* [17]. Similarly, three were infected with *Enc. intestinalis* and four with *Ent. bieneusi* of 70 healthy individuals in Slovakia [18].

In the Republic of Korea (ROK), little research on microsporidia has been reported. In 2011, seven cases of *Enc. intestinalis* were detected among 139 patients with diarrhea [19]. However, no human *Ent. bieneusi* infections have been reported in the ROK, despite infections in wild animals (e.g., water deer and raccoon dogs) and livestock (pigs and calves) [20,21,22,23]. The purpose of this study was to investigate the prevalence of *Enc. intestinalis* and *Ent. bieneusi* and identify their genotypes in the ROK.

## 2. Results

### 2.1. Prevalence of Enc. intestinalis and Ent. bieneusi

Among the 1241 stool samples examined in this study, 24 were positive for *Enc. intestinalis* and one for *Ent. bieneusi* (Table 1). We analyzed other divisions to check for co-infections with bacteria and viruses that could cause diarrhea. Of the 24 *Enc. intestinalis* cases, only one patient was confirmed to be co-infected with *Staphylococcus aureus*. The single patient infected with *Ent. bieneusi* was confirmed to be co-infected with *Campylobacter* spp. Additional results on other parasites, bacteria, and viruses detected during this study will be published later.

*Enc. intestinalis* was detected in 12 male (12/648, 1.9%) and 12 female (12/593, 2.0%) patients. In contrast, *Ent. bieneusi* was detected in only a single male patient (1/648, 0.2%). *Enc. intestinalis* infection was not associated with sex (*p* = 0.8, odds ratio = 0.91, 95% confidence interval = 0.41–2.05). The prevalence of *Enc. intestinalis* was highest in the 30–39 age group (6.5%), followed by the 20–29 (4.7%) and 10–19 (3.5%) age groups. *Ent. bieneusi* was detected in the 10–19 age group, with a prevalence of 1.2% (Table 1).

### 2.2. Sequence Analysis of Enc. intestinalis and Ent. bieneusi Using Small Subunit Ribosomal RNA

*Enc. intestinalis* and *Ent. bieneusi* DNA fragments were sequenced using both the forward and reverse primers Mic C and Mic D. All 24 cases of *Enc. intestinalis* (KDCA 1–24) showed 100% similarity with the small subunit ribosomal RNA (SSU rRNA) gene and were clustered with previously reported isolates (Accession numbers: KM058742, DQ453122, and JF932507), based on their molecular phylogenies. The SSU rRNA sequence of the single *Ent. bieneusi* case (KDCA 25) was also clustered with previously reported isolates (Accession numbers: MG976584, MH027470, and KJ019869) (Figure 1).

### 2.3. Genotype of Ent. bieneusi

Based on the sequencing analysis of the internal transcriptional spacer (ITS) gene, we detected one *Ent. bieneusi* genotype, Korea-WL2. The KDCA25 *Ent. bieneusi* isolate obtained in the present study was identical to LC436502 from a raccoon dog and LC436503 from a Korean water deer in the ROK [20]. KDCA25 showed one nucleotide substitution at position 230 (T/G) compared to existing genotype D sequences (Figure 2). Phylogenetic analysis was carried out to understand the genetic relationships among the *Ent. bieneusi* genotypes. A neighbor-joining tree was constructed using the *Ent. bieneusi* ITS nucleotide sequences from humans and domestic animals. KDCA25 was clustered into Group 1, the human pathogenic group (Figure 3).

## 3. Discussion

*Ent. bieneusi* and *Enc. intestinalis* are both common microsporidia species responsible for gastrointestinal diseases in humans [24]. *Ent. bieneusi* is the most common microsporidial cause of intestinal diseases [25]. However, *Enc. intestinalis* showed a higher infection rate than *Ent. bieneusi* in the ROK. This could have two possible explanations. First, the detection rate depends on patient condition. *Ent. bieneusi* infection accounts for 30–51% of all cases of diarrhea in immunocompromised patients [26]. Previous studies on *Enc. intestinalis* and *Ent. bieneusi* focused on immunocompromised patients. Although this study focused on immunocompetent patients with diarrhea, it is necessary to include immunocompromised patients as a control group in future studies. Second, developed countries have reported gradual decreases in the prevalence and occurrence of *Ent. bieneusi* in immunocompromised patients due to the use of antiretroviral therapies and improved hygiene [27].

In 2015, Kim et al. [19] reported a 5% (7/139) prevalence of *Enc. intestinalis* in patients with diarhhea in the ROK; however, we found a prevalence of 2% (24/1241). In this study, the infection rate was high in the 20–29 (2/43, 4.7%) and 30–39 (3/46, 6.5%) age groups. However, in previous studies, the infection rates were high in the 11–20 (4/15, 26.7%) and 31–40 (1/5, 20%) age groups. In addition, no infection rate was reported for the 21–30 age group by Kim et al. [19] as it contained only one sample, while 20–29 group had the second-highest prevalence in this study. We believe that sampling differences could have affected the low prevalence in this study. As no data related to the infection sources or other clinical information are available for the positive cases in the previous study, it is difficult to explain the differences between our results and those of the previous study.

The ITS domain of rRNA genes is the only known polymorphic marker of *Ent. bieneusi*. At least 474 ITS genotypes have been discovered, which can be systematically classified into 11 groups (Groups 1–11), with Group 1 and 2 being the major groups [28,29]. At least 38 ITS genotypes have been found in the ROK (including genotype D, H, I, J, BEB2, BEB4, BEB8, CAF1, CEbA, CEbB, CEbC, CEbD, CEbE, CEbF, EBITS3, EBITS4, EBITS5, EbpC, KBAT1, KBAT2, KBAT3, KBAT4, KCALF1, KCALF2, KBEB5, KWB1, KWB2, KWB3, KWB4, WL1, WL2, WL3, WL4, WL5, WL6, PigEBITS9, Type IV, and Peru2) and they belong to Group 1 and Group 2 [20,21,22,23,30,31,32]. The genotypes in Group 1 have a wide host range, including humans and numerous mammals, indicating low host specificity and possibility for zoonotic or cross-species transmission [33]. According to previous studies, the genotypes of Group 2 are specific to cattle [34], and those of Group 3 and 4 are specific to muskrat and raccoons [35]. Group 5 also infects humans, including genotype CAF4, which was found in HIV-positive and HIV-negative patients in Gabon and Cameroon [36]. Group 6 contains genotypes found in urban wastewater in China [37], while Group 7 genotypes have been found in HIV-positive patients in Nigeria [38]. Groups 8–11 consist mainly of animal host genotypes [29,39].

In this study, the *Ent. bieneusi* isolate (KDCA 25) belonged to Group 1. The sequence matched with those previously reported from domestic wildlife [19], suggesting a possible infection through fecal-oral route via contaminated water, the environment, or food from domestic wildlife [35]. There have been no previously reported cases of *Ent. bieneusi* in the ROK, except for domestic and wild animals. Previous studies on *Ent. bieneusi* infection have shown a total infection rate of 45.2% (71/157) in domestic animals [3] and 18% (53/314) in native calves [21]. Other studies have reported infection rates of 16% (38/237) in pigs with diarrhea [22] and 14.9% (80/538) in cattle [23], 1.9% (4/210) in bat feces [30], 2.6% (15/502) in wild boars [31], and 8.3% (15/180) in milk specimens from cows [32]. As previously reported, the possibility of human infection is suggested by the continuous reports of *Ent. bieneusi* infections in domestic wildlife and livestock. Microsporidian spores remain viable in water after desiccation following incubation at various temperatures, suggesting that indirect zoonotic transmission of microsporidia between animals and humans could occur through exposure to contaminated water, food, or aerosols [6,40,41,42].

In this study, we detected *Ent. bieneusi* and *Enc. intestinalis* in domestic patients with diarrhea, providing the first evidence that *Ent. bieneusi* exists in humans in addition to animals in the ROK. Therefore, continuous monitoring for *Enc. intestinalis* and *Ent. bieneusi* is necessary for patients with acute diarrhea. In additon, epidemiologic analysis is needed to understand the infection pathways of *Enc. intestinalis* and *Ent. bieneusi*.

## 4. Materials and Methods

### 4.1. Fecal Sample Collection and DNA Extraction

A total of 1241 stool samples from patients with diarrhea were collected through the Enteric Pathogens Active Surveillance Network (Enter-Net) of the Korea Disease Control and Prevention Agency (KDCA) in 2020. The samples were examined for bacteria, viruses, and parasitic protozoa to determine the cause of diarrhea.

Total DNA was extracted from 400 mg stool samples using the Fast DNA SPIN kit for Soil (MP Biomedicals, Solon, OH, USA) according to the manufacturer’s protocol. The extracted DNA was stored at −20 °C until PCR analysis.

### 4.2. PCR Amplification

*Enc. intestinalis* and *Ent. bieneusi* were detected using nested PCR, targeting the SSU rRNA [43]. The primers used for PCR amplification are listed in Table 1. The cycling conditions were as follows: the primary cycle consisted of 94 °C for 5 min; 35 cycles of 94 °C for 30 s, 53 °C for 30 s, and 72 °C for 90 s; followed by 72 °C for 10 min; and termination at 4 °C. The second cycle consisted of 94 °C for 5 min; 35 cycles of 94 °C for 30 s, 55 °C for 30 s, and 72 °C for 90 s; followed by 72 °C for 10 min; and termination at 4 °C.

For genotyping *Ent. bieneusi*, a 390 bp fragment of the ITS gene was amplified using nested PCR [44]. The primer sequences are listed in Table 1. The cycling conditions for *Ent. bieneusi* were as follows: the primary cycle consisted of 94 °C for 5 min; 35 cycles of 94 °C for 30 s, 57 °C for 40 s, and 72 °C for 40 s; followed by 72 °C for 10 min; and termination at 4 °C. The second cycle consisted of 94 °C for 5 min; 35 cycles of 94 °C for 30 s, 55 °C for 30 s, and 72 °C for 40 s; followed by 72 °C for 10 min; and termination at 4 °C.

### 4.3. Nucleotide Sequencing and Phylogenetic Analysis

The amplified *Enc. intestinalis* and *Ent. bieneusi* DNA fragments were sequenced using both forward and reverse primers (Table 2). The resulting nucleotide sequences were subjected to BLAST searches (https://blast.ncbi.nlm.nih.gov/Blast.cgi accessed on 13 October 2021) using the data available on GenBank (http://www.ncbinlm.nih.gov/genbank/ accessed on 13 October 2021). Multiple alignments were conducted by ClustalW using BioEdit version 7.2.5 (Ibis Therapeutics Inc., Carlsbad, CA, USA). To determine the similarity and difference rates between the sequences, the MegaAlign program (DNASTAR, Madison, WI, USA) was employed. Finally, a phylogenetic analysis was performed with MEGA software (version 5.02) using the maximum parsimony algorithm with a Kimura two-parameter model assessed using bootstrap analysis with 1000 replications. Each sequence was identified by its accession number, host, origin, and genotype designation.

### 4.4. Statistical Analysis

Fisher’s exact test was used to assess the association between *Enc. intestinalis* test positivity and factors such as sex and age group. Odds ratios and 95% confidence intervals were used to measure univariate associations. In this study, *p*-values < 0.05 were considered statistically significant. All statistical analyses were performed using SPSS Statistics 23.0 (International Business Machines Corporation, New York, NY, USA).

## Figures and Tables

**Figure 1 pathogens-10-01424-f001:**
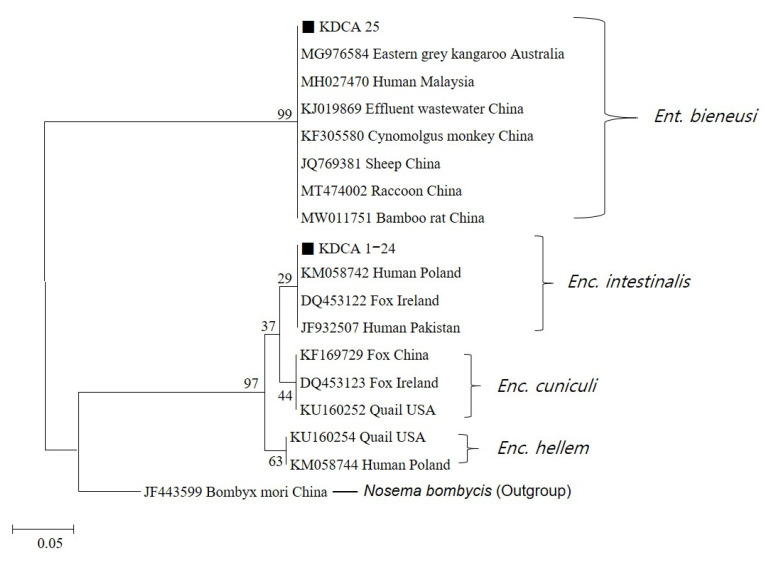
Phylogenetic tree of *Encephalitozoon* spp. and *Enterocytozoon bieneusi* isolates and selected accessions from GenBank, based on small subunit ribosomal RNA gene fragment sequences. The phylogenetic tree was constructed using nucleotide sequence alignments, with the Kimura two-parameter algorithm as the distance method and neighbor-joining as the tree composition method. The black squares indicate the known genotypes identified in this study. The sequence of *Nosema bombycis* was used as the outgroup.

**Figure 2 pathogens-10-01424-f002:**
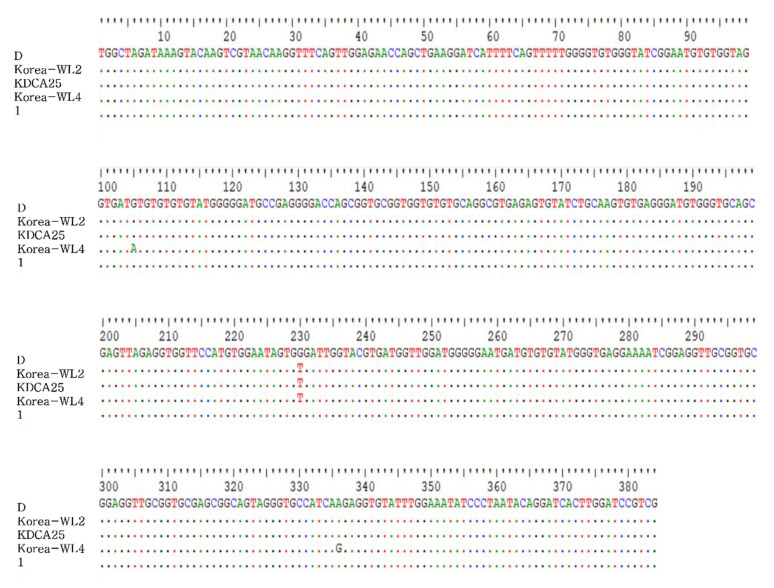
Sequence based on internal transcriptional spacer (ITS) sequences. Position of nucleotide changes within *Enterocytozoon*
*bieneusi* genotype D-related sequences compared to that in MK696083 (D), LC436503 (Korea WL-2), LC436512 (Korea WL-4), and MN922367(1). Each sequence is identified by its genotype designation. Sequence differences of ITS sequences obtained in this study are shown. “.” indicates an identical nucleotide to MK696083 (D).

**Figure 3 pathogens-10-01424-f003:**
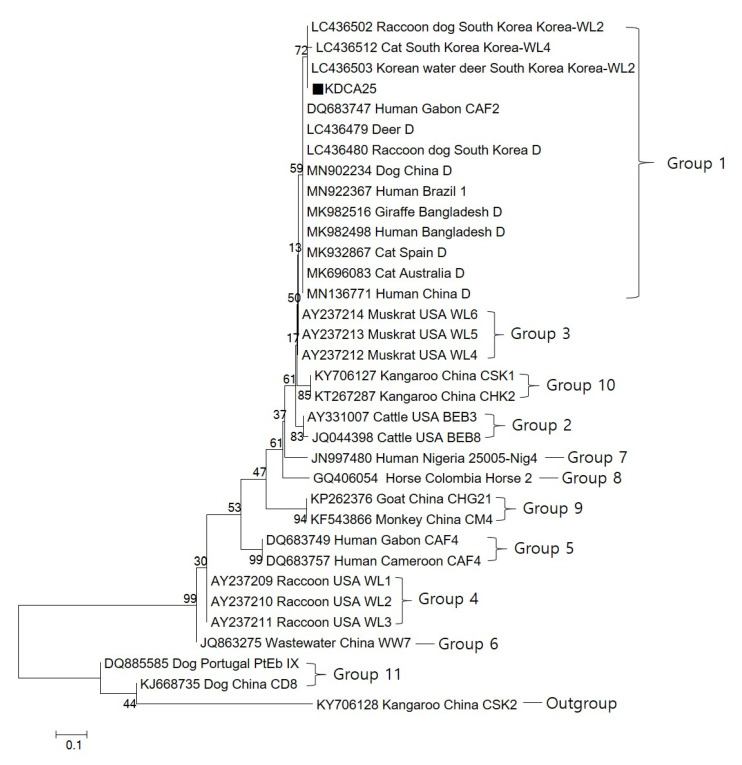
Phylogenetic tree based on neighbor-joining analysis of internal transcriptional spacer (ITS) sequences. The phylogenetic relationships between the *Enterocytozoon bieneusi* genotype determined in this study and previously reported genotypes from GenBank were inferred by neighbor-joining analysis of ITS sequences, based on Kimura two-parameter model genetic distances. The numbers on the branches are percentage bootstrapping values from 1000 replicates. Each sequence is identified by its accession number, host, origin, and genotype designation. The black square indicates the known genotype identified in this study.

**Table 1 pathogens-10-01424-t001:** *Encephalitozoon intestinalis* and *Enterocytozoon bieneusi* infection rates.

Protozoa Parasite	Total	*Enc. intestinalis* (%)	*Ent. bieneusi* (%)
**Number of Samples**	1241	24 (2.0%)	1 (0.1%)
**Sex**	**Total**	***Enc. intestinalis* (%)**	***Ent. bieneusi* (%)**
Male	648	12 (1.9%)	1 (0.2%)
Female	593	12 (2.0%)	−
**Age Group (Years)**	**Total**	***Enc. intestinalis* (%)**	***Ent. bieneusi* (%)**
0–9	281	3 (1.1%)	−
10–19	86	3 (3.5%)	1 (1.2%)
20–29	43	2 (4.7%)	−
30–39	46	3 (6.5%)	−
40–49	77	2 (2.6%)	−
50–59	144	1 (0.7%)	−
60–69	202	5 (2.5%)	−
≥70	362	5 (1.4%)	−
**Total**	1241	24 (2.0%)	1 (0.1%)

**Table 2 pathogens-10-01424-t002:** List of PCR primers used in this study.

Species	Primer		Sequence (5′→3′)	Diagnostic Size
***Microsporidia* spp.**	1st	Mic A	GGAGCCTGAGAGATGGCT	644 bp
		Mic E	AACGGCCATGCACCAC	
	2nd	Mic C	GGTGCCAGCAGCCGCGG	420 bp
		Mic D	GCACAATCCACTCCT	
** *Ent. bieneusi* **	1st	EBITS3	GGTCATAGGGATGAAGAG	435 bp
		EBITS4	TTCGAGTTCTTTCGCGCTC	
	2nd	EBITS1	GCTCTGAATATCTATGGCT	390 bp
		EBITS2.4	ATCGCCGACGGATCCAAGTG	

## Data Availability

The datasets generated during and/or analysed during the current study are available from the corresponding author on reasonable request.

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
