# Peer review of "First Identification and Genotyping of Enterocytozoon bieneusi and Prevalence of Encephalitozoon intestinalis in Patients with Acute Diarrhea in the Republic of Korea"

_pathogens, 2021, doi:10.3390/pathogens10111424_

Round 1
Reviewer 1 Report
Dear authors,
Thanks for this interesting manuscript.
Please revise the sceuntific name of parasites to be italic in the whole manuscript.
Also, please the same of bacteria names Staphylococcus aeurus and Campylobacter, it should be italic.
In the result section, you should add sentence that other results for other parasites, bacteria, and viruses detected during the study will be published later.
Thanks.
Regards
Reviewer 2 Report
In the manuscript entitled “First identification and genotyping of Enterocytozoon bieneusi and prevalence of Encephalitozoon intestinalis in the Republic of Korea patients with acute diarrhea” the authors have discussed the identification and characterization of the above-mentioned Microsporidia in the Republic of Korea. Below are the suggestions to improve the manuscript.
- Apart from PCR amplification and genotyping, did the authors use other tests to corroborate their finding? I think it would be a good idea to confirm their results.
- Line# 131-132: The ITS domain of rRNA genes is the only known polymorphic marker of bieneusi. The authors should italicize the Microsporidia names throughout the manuscript.
- Why is it that the immunocompetent patients are more susceptible to infections from Encephalitozoon intestinalis? However, the immunocmpromised patients are more susceptible to Enterocytozoon bieneusi. The authors should discuss in detail about the possible mechanisms in the discussion.
